# Propolis Ameliorates Alcohol-Induced Depressive Symptoms in C57BL/6J Mice by Regulating Intestinal Mucosal Barrier Function and Inflammatory Reaction

**DOI:** 10.3390/nu14061213

**Published:** 2022-03-12

**Authors:** Peng Wang, Peiyu Guo, Yanhui Wang, Xiangyun Teng, Huaqi Zhang, Lirui Sun, Meilan Xue, Hui Liang

**Affiliations:** 1The Institute of Human Nutrition, Qingdao University, Ning Xia Road 308, Qingdao 266071, China; wpeng@qdu.edu.cn (P.W.); gpy610707@163.com (P.G.); 17863810524@163.com (Y.W.); txy20180209@126.com (X.T.); huaqi_erin@163.com (H.Z.); yzslr1222@foxmail.com (L.S.); 2Department of Biochemistry and Molecular Biology, Basic Medical College, Qingdao University of Medicine, Ningxia Road 308, Qingdao 266071, China; snowml@126.com

**Keywords:** propolis, alcohol, depression, brain gut dysfunction

## Abstract

Accumulating evidence points to a critical role of the brain gut axis as an important paradigm for many central nervous system diseases. Recent studies suggest that propolis has obvious neuroprotective properties and functionality in regulating intestinal bacteria flora, hinting at a potential key effect at both terminals of this axis regulation. However, currently no clear evidence confirms the effects of propolis on alcohol-induced depression. Here, we establish an alcoholic depression model with C57BL/6J mice and demonstrate that treatment with propolis protects against alcohol-induced depressive symptoms by behavioral tests. In addition, propolis attenuates the injury of nerve cells in the hippocampal region and restores the serum levels of brain-derived neurotrophic factor (BDNF) and dopamine (DA) in mice with alcohol-induced depression. Pathology and biotin tracer assays show that propolis repairs the intestinal leakage caused by alcohol. Additionally, propolis treatment increases the expression levels of intestinal intercellular tight junctions’ (TJs’) structural proteins Claudin-1, Occludin and zona occludens-1 (ZO-1), as well as the activation state of the liver kinase B1/AMP-activated protein kinase (LKB1/AMPK) signaling pathway, which is closely related to the intestinal permeability. Furthermore, propolis can reduce the levels of pro-inflammatory, lipopolysaccharide (LPS) and fatty-acid-binding protein 2 (FABP2), suggesting the significance of the inflammatory response in alcoholic depression. Collectively, our findings indicate that propolis exerted an improving effect on alcohol-induced depressive symptoms by ameliorating brain gut dysfunction.

## 1. Introduction

Depression, characterized by disrupted mood, along with an array of symptoms of the emotional, motivational, cognitive and physiological domains, is the leading cause of psychiatric disability around the world [1,2]. According to a report of World Health Organization (WHO), it is estimated that more than 300 million people of all ages suffer from depression, resulting in significant economic and emotional strain on society [3]. Over the past 40 years of research, numerous risk causes for depression have been identified, including both genetic and environmental factors [4,5]. Of note, alcohol abuse is frequently linked to depression and even serves as a serious risk factor for the development of despondent mental diseases [6]. In addition, increasing amounts of evidences show that comorbid depression and problematic alcohol use often co-occur and tremendously enhance suicide-related events [7,8,9,10,11].

Although many hypotheses have been proposed to explain the biological mechanism of alcoholic depression, such as short-chain fatty acids [12], intestinal microbiota [13,14] and neuropeptide systems [15], key information about its regulation remains ambiguous [16]. In recent years, more and more studies reveal the important effect of the brain gut axis on the regulation of depression, which was able to establish pathways of bidirectional communication by a variety of media, including the immune system [13,17]. Excessive alcohol consumption has been suggested to serve as a modulator of the inflammatory response. Alcohol abuse can destroy the integrity of the intestinal barrier and inhibit the expression of tightly connected related proteins, leading to enhanced intestinal mucosal permeability and endotoxin intestinal leakage, as well as an immune system response of the central nervous system (CNS) [18,19]. Therefore, the inflammatory response is also an important marker of intestinal barrier dysfunction. The intestinal leakage hypothesis is also considered to be an important mechanism to explain the depression caused by inflammation [20]. Heightened inflammation characterizes a series of immune system activation, and is likely to transmit inflammatory signals to the brain, leading to brain damage and neurological disorders, each of which also feature as an elevated risk for depression [21]. Moreover, several studies show that depression and inflammation fuel one another [22]. Patients with major depression exhibit significant inflammatory responses, including the increased expression of pro-inflammatory cytokines such as interleukin 6 (IL-6), interleukin 18 (IL-18) and tumor necrosis factor α (TNF-α) [23,24]. Therefore, the effective improvement and regulation of intestinal mucosal barrier function might be a considerable therapeutic approach for alcoholic depression through reducing the inflammatory reaction.

Nutritional intervention with functional foods is an alternative therapeutic strategy for the treatment of mental illness [25,26]. Propolis is a natural gelatinous solid substance collected by honeybees from plant exudates, beeswax and bee secretions, and is widely used as a dietary supplement due to its wide spectrum of biological activities [27]. Propolis has obvious anti-inflammatory function, with evidence indicating that it may be a candidate for the treatment of inflammatory bowel disease (IBD) [28,29]. Recent animal experiments have also shown that propolis was able to modulate gut microbiota and improve the intestinal mucosal barrier [30,31,32]. In addition, studies have shown that propolis possessed a neuroprotective effect, therefore being able to act against the brain dysfunction induced by multiple neurotoxins [33,34]. However, currently no clear evidence confirms the effects of propolis on alcohol-induced depression.

In this study, we aimed to investigate the protective effects of propolis on alcohol-induced depressive symptoms in C57BL/6J mice. Moreover, we explored the deeper functional mechanism of propolis on the brain gut dysfunction caused by intestinal mucosal barrier injury and systemic inflammatory responses to understand the potential value of anti-alcoholic depression.

## 2. Materials and Methods

### 2.1. Propolis Preparation

The propolis used in the present study was extracted and purified from raw propolis collected from Shandong province, China, and the botanical origin was poplar (*Populus* sp.). The preparation of hydroethanolic extracts of propolis was carried out as described by Zhang H, et al. [33] with slight modification. Briefly, crude propolis was frozen at −40 °C and ground into powder. Then, propolis samples of 20 g were treated with 400 mL of 75% (*v*/*v*) aqueous ethanol solvent for 3 h, followed by being placed in a constant temperature ultrasonic extractor (JRA-2000T, Jieruian, Wuxi, China) and sonicated for 5 h at 50 °C. After that, the sample was filtered off through a 0.45 μm microporous membrane and dried into a solid substance. Propolis extract was dissolved with 10% ethanol and adjusted to the required concentration (weight to volume), followed by being stored at 4 °C in no-light conditions and warmed to room temperature until use. High-performance liquid chromatography (HPLC) was performed to determine the total flavonoids content, as described in our previous study [30].

### 2.2. Animals, Experimental Models and Intervention Strategy

In this study, 10-week-old male C57BL/6J mice, weighing 20 ± 2 g, were obtained from Vital River Laboratory (Beijing, China). The healthy mice were kept in the specific-pathogen-free (SPF) animal center of Qingdao University, at a set relative humidity of 50–60% and a temperature of 21–23 °C, precisely controlled by intelligent air-conditioning equipment. During the study, the mice were fed with standard rodent chow and water, and acclimated to the environment for a week to make the life characteristics tend to be stable and consistent before the start of experiments. The protocols for animal experiments were approved by the Animal Care and Use Committee of the Medical College, Qingdao University, and strictly conducted in accordance with the guidelines of laboratory animals of the National Institutes of Health.

The establishment of alcohol-induced depression models and an intervention strategy was carried out according to our previous report by Jiang et al. [34] with slight modification. Briefly, the mice were randomly separated into the three groups (*n* = 10 in each group): control, model, and propolis groups. Subsequently, each mouse was earmarked, weighed and housed individually. After one week of adaptive feeding, the experiment began and lasted for 10 weeks. From Monday to Thursday, the mice in the model and propolis groups were provided with 15% alcohol solution (*v*/*v*) from 4:00 p.m. to 8:00 a.m. the next day, and fresh clean water was provided the rest of the time. Afterwards, all the liquid supply was cut off on Friday and Saturday, and the fresh clean water supply was restored on Sunday. In addition, mice of the control and model groups received 0.2 mL/d of soybean oil using intragastric administration at 12:00 a.m. each day, and the Propolis group was given 120 mg/kg of propolis by gavage daily.

### 2.3. Sucrose Preference Test (SPT)

Before the SPT experiment, the mice were adapted to the sucrose solution in a quiet environment for 48 h separately. Two identical water bottles containing the same amount (100 mL) of 1% sucrose solution were placed in each cage for 24 h, followed by replacing one of the bottles with one containing 100 mL of fresh clean water for another 24 h to avoid sucrose neophobia. All the mice took chow freely in the experimental period. After this, the mice were transferred to the environment of the SPT apparatus for apparatus adaptation for 24 h. The method of eating and drinking was the same as above. Then, the mice were deprived of both food and water for 24 h after adaptive training, and the SPT experiment began. Immediately after deprivation, two bottles containing 100 mL of 1% sucrose water or 100 mL of 1% fresh clean water were weighed in advance and randomly placed in each cage. In order to avoid habitual drinking water, the positions (left right) of the sucrose solution and water bottle were switched every 0.5 and 6 h, and the weight was recorded at intervals of 1 h and 12 h. The consumption of sucrose and water was measured by comparing the weights of the bottles before and after the formal SPT experiment. The sucrose preference was calculated as the following formula: Sugar preference value = sugar consumption/(sugar consumption + water consumption) × 100%. After the test, all the mice were returned to group housing, with free access to chow and water. Notably, in consideration of the effect of circadian rhythms on the drinking of mice, the formal SPT experiment was carried out during the night.

### 2.4. Open-Field Test (OFT)

A square box (50 cm × 50 cm × 50 cm) with a black inner wall color was used for the OFT experiment. The test arena was divided into a central square (25 cm × 25 cm) and a peripheral area. On the day of the test, all the mice were transferred to the testing room and placed in their cages for 2 h before the test. During the experiment, a single mouse was placed in a particular corner of the open-field box and allowed to explore freely for 3-min periods in a dimly illuminated and quiet room. A video camera (Handycam, Tokyo, Japan) and video-tracking system (SMARTV3.0, Panlab, Barcelona, Spain) were used to record the behavioral responses, including the total distance travelled and the locomotion time in the central square area as well as the peripheral area. After each test, the mouse was moved out and seventy percent alcohol was used to clean the box to eliminate olfactory cues from the previously tested mouse.

### 2.5. Elevated Plus Maze (EPM)

An elevated plus maze apparatus was used for the EPM experiment. The test apparatus comprised a cross-shaped platform, painted matte white, which was elevated 80 cm above the floor. It consisted of two opposite closed arms (flanked by 30 cm opaque walls) and two opposite open arms (without walls), and was linked by a neutral zone in the center area. All the arms were 50 cm long and 10 cm wide, and the central zone was 10 cm × 10 cm, allowing mice to move freely into each area of the maze. The test was performed in a quiet room from 9:00 a.m. to 1:00 p.m., to avoid the possible effects of daily changes in plasma corticosterone levels. During the test, mice in each group were picked up by the tail and placed in the center area, facing one open arm, and were allowed to explore freely in the maze for alternating 5-min periods. A video camera (Handycam, Tokyo, Japan) and video-tracking system (SMARTV3.0, Panlab, Barcelona, Spain) were used to record the behavioral responses, including the frequency of entries, distance travelled and the locomotion time in the open arms. After each test, the EPM apparatus was cleaned with seventy percent alcohol, as described above.

### 2.6. Forced Swim Test (FST)

Before the FST experiment, the mice, in their cages, were transported to the testing environment for least 30 min prior to beginning the behavioral testing. The FST experiment was performed via a PLEXIGLAS cylinder (25 cm in height × 15 cm in diameter) filled with water (23 ± 1 °C). The depth of the water was adjusted so that the mice’s feet could not touch the bottom of the container (approximately 15 cm). Generally, after entering the water, mice swim hard to seek an escape route, and then stop swimming and instead float on the surface of the water passively. In particular, mice with depressive symptoms give up the struggle faster and solely do what is necessary just to keep their head out of the water, appearing immobile. During the test, mice in each group were forced to swim for a period of 6 min, and behavioral responses from the last 4 min were recorded by using a video camera (Handycam, Tokyo, Japan); the immobility time was measured by using SMART V3.0 software (Panlab, Barcelona, Spain) as we described before [34]. After each test, the mouse was taken out of the water and dried on a heating pad, and the water in the container was replaced to eliminate olfactory cues from the previously tested mouse.

### 2.7. Preparation of Specimens

After behavioral experiments, the mice were in narcotism by injecting pentobarbital sodium (40 mg/kg) into their cavum abdominis, and their blood samples were collected from the eyeball before they were sacrificed by the dislocation of their cervical vertebra. After being coagulated, the blood was centrifuged at 956× *g* for 5 min to obtain the serum specimens. The mice were decapitated, which was followed by the hippocampal tissues in the brain being quickly dissected for removal and becoming accordingly fixed in 10% paraformaldehyde solution for histopathological analysis. The intestinal tissue from each group was rapidly dissected, segmented, cut into blocks and fixed with 2.5% glutaraldehyde solution or 10% paraformaldehyde solution for transmission electron microscopy or histopathological analysis, respectively. The remaining intestinal tissue and spleen were frozen in liquid nitrogen and stored immediately at −80 °C until use.

### 2.8. Histopathology

After the mice were sacrificed, a small portion of the hippocampus and intestinal tissue was quickly excised and fixed in 10% paraformaldehyde solution for 24 h, immediately. Then, the samples were dehydrated with serial alcohol (50%, 70%, 90% and absolute), embedded in paraffin and sectioned to 5 μm thickness with an RM 2135 rotary microtome (Leica, Wetzlar, Germany). Subsequently, the slices were dewaxed in xylene for 10 min, followed by immersion in absolute ethanol for 10 min, and then rehydrated in 90%, 70% and 50% ethanol. After rinsing with distilled water for 5 min, the sample was dyed with hematoxylin and eosin (HE) in strict accordance with the standard procedure. The observation and photography were carried out using a microscope system (Olympus, Tokyo, Japan). The denatured cell index (DCI = the number of denatured cells/the number of total cells) was analyzed to evaluate the injury severity of the hippocampus.

The jejunum and colonic tissues used for transmission electron microscopy, were cut into 1 mm × 2 mm sections and fixed with 2.5% glutaraldehyde solution at 4 °C for more than 24 h. Subsequently, the samples were refixed with 1% osmium tetroxide for 1.5 h at 4 °C after being washed using a phosphate-buffered saline (PBS) (pH 7.4), followed by being dehydrated in graded acetone (30–100%) and deposited in the epoxy resin. Then, the ultrathin sections (70 nm) were cut on an Ultracut E ultramicrotome (Reichert-Jung, Vienna, Austria), collected on copper grids and stained with 3% uranyl acetate and lead citrate, respectively. Finally, after the sample was rinsed with double-steamed water, the ultrastructure of the jejunal and colonic sections were examined using a JEM-1200EX transmission electron microscope (JEOL, Tokyo, Japan).

### 2.9. ELISA (Enzyme-Linked Immunosorbent Assays)

The spleen tissue was homogenized in PBS solution (pH 7.4) and centrifuged at 11,000× *g* for 20 min, 4 °C. The supernatant was collected into a new tube and the total protein concentrations were quantified, respectively, using a bicinchoninic acid (BCA) protein assay kit (CWBIO, Beijing, China). Subsequently, the protein levels of TNF-α, IL-6 and IL-18 in spleen homogenates and the levels of brain-derived neurotrophic factor (BDNF), dopamine (DA), 5-hydroxytryptamine (5-HT), fatty-acid-binding protein 2 (FABP2) and lipopolysaccharide (LPS) in serum were measured by using ELISA kits (Nanjing Jiancheng Biological Engineering Research Institute, Nanjing, China), respectively.

### 2.10. Immunofluorescence

The jejunum and colonic tissues samples were sectioned to 5 μm thickness with an RM 2135 rotary microtome (Leica, Wetzlar, Germany), followed by xylene dewaxing and gradient ethanol hydration as described above. Subsequently, the sections were preincubated with 4% bovine serum albumin (BSA) in PBS (pH 7.4) at room temperature for 30 min, and then incubated with a specific anti-Claudin-1 primary antibody (1:800) (Cell Signaling Technology, Danvers, USA) and goat anti-zona occludens-1 (ZO-1) primary antibody (1:200) (Cell Signaling Technology, Danvers, USA), overnight at 4 °C, respectively. Then, the sections were washed 3 times with PBS (pH 7.4) and incubated further at room temperature for 60 min with a corresponding secondary antibody. The nucleus was stained with 4′,6-diamidino-2-phenylindole (DAPI). The distributions of Claudin-1 and ZO-1 in intestinal epithelial cells were observed and photographed using a Zeiss microscope (Zeiss, Oberkochen, Germany).

### 2.11. Western Blotting

The jejunum and colon tissues were homogenized with Tissue Protein Extraction Reagent (Pierce Biotechnology, Danvers, MA, USA)-containing protease inhibitors and phosphatase inhibitors, and centrifuged at 11,000× *g* for 20 min, at 4 °C. The supernatant was collected into a new tube; the total protein concentrations were quantified, respectively, using a BCA protein assay kit (CWBIO, Beijing, China). Equal amounts of protein extracts from each group were subjected to separation on 12% sodium dodecyl sulfate-polyacrylamide (SDS-PAGE) gels and transferred onto polyvinylidene difluoride membranes (Millipore, Billerica, MA, USA). Subsequently, the membranes transferred with protein were blocked using a Tris-buffered saline (TBS) buffer containing 5% BSA for 30 min at room temperature, and then incubated with specific primary antibodies, including anti-Ras homolog gene family member A (RhoA) (1:1000), Claudin-1 (1:1000), Occludin (1:1000), ZO-1 (1:1000), liver kinase B1 (LKB1) (1:1000), p-LKB1 (1:1000), AMP-activated protein kinase (AMPK) (1:1000), p-AMPK (1:1000) and β-actin (1:1000) (Cell Signaling Technology, Danvers, MA, USA), overnight at 4 °C, respectively. β-actin was chosen as the reference for internal standardization. After being washed with the TBS buffer five times, the membranes were incubated with the corresponding secondary antibodies (Zymed Laboratories, San Francisco, CA, USA) for 40 min under room temperature. Then, after being washed with the TBS buffer five times, protein bands were detected by an enhanced chemiluminescence (ECL) Western-blotting kit (BioVision, Milpitas, CA, USA). Quantification of the protein band intensity was performed using Image-J software, version 1.46r.

### 2.12. Tracer Experiment

The tracer experiment was performed to determine the structural integrity of the intestinal barrier, as described in our previous study [35]. EZ-link Sulfo-NHS-Biotin (Pierce Chemical, Rockford, IL, USA) was used as a molecular tracer, which was diluted to a concentration of 2 mg/mL with PBS (pH 7.4) plus 1 mM CaCl_2_ before use. Briefly, a small section (about 2 cm) of jejunum and colon was isolated immediately after the mice were sacrificed, and placed in PBS (pH 7.4) at 37 °C. Then, the prepared biotin was added to the intestinal cavity gently, followed by ligation at both ends. The intestinal tissue samples were incubated at room temperature for 5 min, and then fixed with 4% paraformaldehyde (PFA) in PBS (pH 7.4) for 3 h. After being washed four times (5 min each) in ice-cold PBS (pH 7.4), the fixed tissue was cut into 5 μm slide sections via freezing microtome (Leica, Wetzlar, Germany) and then incubated with a 1:800 dilution of Alexa-Fluor-488-conjugated streptavidin for 30 min under the light-proof conditions. The distributions of biotin in intestinal epithelial cells were observed and photographed using a Zeiss microscope (Zeiss, Oberkochen, Germany).

### 2.13. Statistical Analysis

All data in this study were analyzed using SPSS22.0 statistical software (SPSS, Chicago, IL, USA) and GraphPad Prism 8 (GraphPad, San Diego, CA, USA). Quantitative data were presented as the mean ± SD, and a one-way ANOVA was performed to assess the differences between the groups. *p* < 0.05 was considered as significantly different statistically.

## 3. Results

### 3.1. Body Weight and Daily Food Consumption

Mice in each group were periodically fed on diets, and the body weight and food consumption were monitored for 10 weeks. As shown in Figure 1A, the changes in body weight were influenced by the addition of alcohol to the diet, especially from the fifth week onwards (*p* < 0.05). However, after the treatment with propolis, the alcohol-induced reduction in body weight was alleviated, and there was a significant difference between the model group and the propolis group at the end of the intervention (*p* < 0.05). The evidence suggested that alcohol was able to affect food intake via pharmacological actions and excessive energy production by alcohol metabolism [36]. As shown in Figure 1B, the daily food consumption of mice in the model group was generally lower than of those in the control groups, and there were significant differences at weeks four, seven and ten (*p* < 0.05). Likewise, propolis treatment showed an effective improvement in food consumption during the last week of the experiment (*p* < 0.05).

### 3.2. Propolis Ameliorates Alcohol-Induced Behavioral Deficits

The effect of propolis on alcohol-induced behavioral changes in mice was detected by various behavioral experiments. After alcohol exposure, the sucrose preference of model group mice was significantly decreased at 12 h compared with the control group, and the supplementation of propolis was able to alleviate this preference change (*p* < 0.05; Figure 2A). In the open-field test, although no significant difference was found in the total movement distance among the three groups, the actions trajectory and the time spent of model group mice significantly reduced in the central square zone compared with those of the control and propolis groups (*p* < 0.05; Figure 2C,D,F). Similarly, the total distance, the locomotion time and the number of entries in the open arms of the EPM were significantly reduced in the model group compared with those of the control and propolis groups (*p* < 0.05; Figure 2E,G,H,I). Furthermore, alcohol exposure caused an obvious increase in the immobility time of mice in the forced swim test compared with the levels of the normal control, and propolis treatment significantly attenuated the prolonged immobility time of mice induced by alcohol exposure (*p* < 0.05; Figure 2B). Collectively, these results of behavioral experiments showed that alcohol exposure was able to induce depression-like behavior in mice, and that propolis treatment effectively alleviates these symptoms.

### 3.3. Propolis Treatment Attenuates the Alcohol-Induced Injury of Nerve Cells in the Hippocampal CA3 Region in the Brain

HE staining was conducted to detect the effects of propolis on neuronal damage in the hippocampus of mice treated with alcohol. As shown in Figure 3A(a,a’), the structure of hippocampal neurons from the control group displayed normal morphology, with an orderly cell arrangement and clear nucleolus, while the alcohol exposure caused a certain degree of hippocampal damage. In the model group, the neurons in the hippocampal CA3 region were sparsely arranged and disordered, their cytoplasm was condensed and plenty of cell degeneration as well as necrosis emerged (Figure 3A(b,b’)). However, compared with the model group, the severity of the pathological changes in hippocampal nerve cells was significantly attenuated in the propolis group, with a relatively orderly arrangement, normal layers and a reduced number of denatured cells (Figure 3A(c,c’)). Moreover, the analysis of the denatured cell index (DCI) revealed that the DCI of hippocampus in model mice was higher than that in control mice, while propolis supplementation (treated group) was able to significantly reduce the DCI of hippocampus compared with the alcohol model group (*p* < 0.05; Table 1). These results suggested that propolis shows a protective effect on alcohol-induced injuries of nerve cells in the hippocampal CA3 region.

### 3.4. Propolis Diminishes the Alcohol-Induced Decrease in BDNF and Monoamine Expression in the Serum of Mice

BDNF is a key transducer of antidepressant effects through neuroprotection and neuroregeneration [37,38]. As shown in Figure 3B, an ELISA revealed that alcohol exposure significantly reduced the levels of BDNF in the serum compared with the normal control (*p* < 0.05), whereas propolis treatment was able to reverse the changes induced by alcohol (*p* < 0.05). Monoamines, particularly the DA and 5-HT neuromodulatory systems, play an important role in reward and aversion [39]. Thus, we determined the serum concentration of DA and 5-HT via an ELISA. As shown in Figure 3C, the DA levels of the alcohol model group mice were significantly lower than those of the control group (*p* < 0.05). As expected, propolis treatment led to upregulated DA compared to the model group (*p* < 0.05). However, the levels of 5-HT showed no significant difference among the three groups (Figure 3D; *p* < 0.05). These results suggested that propolis was able to regulate the expression of factors associated with depressive-like behaviors.

### 3.5. Effect of Propolis on the Intestinal Mucosal Barrier in Alcohol Exposed Mice

Alcohol is one of the main factors causing intestinal barrier dysfunction [40,41]. Here, light microscopy and transmission electron microscopy were performed to detect the effect of propolis on intestinal mucosal injuries. HE staining images under a light microscope showed that the different intestinal tissue structures of mice in the control group were intact; the gland and epithelial monolayer columnar cells were arranged regularly; and the villi were smooth, clear and lined in neat rows. However, in the model group, the intestinal tissues presented as being obviously atrophied and collapsed, and the villous structures were damaged, with local epithelial shedding. In addition, the ratio of villus height to crypt height decreased, and a partial oedema was seen in the lamina propria of the jejunum and ileum. After dietary propolis supplementation, the morphological structure of different locations of intestines was improved and the damaged mucosa was restored to some degree, the intestinal wall was complete and the structure was clear (Figure 4A).

The ultrastructure of the intestine was observed using a transmission electron microscope. As shown in Figure 4C, the intercellular junctions between intestinal epithelial cells were intact in the control group, and the intercellular space is clearly visible. After alcohol exposure in the model group, the tight junctions (TJs) were damaged, and the membrane fusion between the intestinal epithelial cells and the gap was blurred and abnormally widened. After propolis treatment in the intervention group, the intercellular junction structure of epithelial cells was improved and the intercellular spaces were narrowed. These results suggested that propolis treatment was able to repair the abnormal morphological and pathological changes of intestinal mucosal epithelial cells caused by alcohol.

### 3.6. Propolis Alleviates the Alcohol-Induced Injury of Intestinal Permeability

Intercellular TJs structures are primarily responsible for the maintenance of the selective permeable barrier, regulating the passage of ions and solutes between cells via multiple protein complexes including Claudin-1, Occludin and ZO-1. To further confirm the structural integrity of TJs, the distribution of Claudin-1 and ZO-1 in the jejunum and colon was detected by the immunofluorescence technique. As shown in Figure 4B, in the control group, Claudin-1 and ZO-1 were highly expressed on the surface of columnar epithelial cells in jejunum and colon that form a continuous barrier. However, after alcohol exposure, the expression levels of Claudin-1 and ZO-1 proteins decreased significantly, while the propolis treatment was able to ameliorate the changes in these two proteins. Consistently, Western blotting also showed that the expression levels of Claudin-1, Occludin and ZO-1 in the jejunum and colon in model group were lower than those in the control and propolis groups (Figure 4E,F; *p* < 0.05). In addition, a biotin tracer assay was carried out to visually assess the permeability of the jejunal and colonic epithelium. As shown in Figure 4D, the signals of biotin fluorescent (green) were restricted in the jejunal and colonic lumen, while partial intestinal leakage appeared in the model group compared with the control group. However, propolis intervention was able to alleviate this symptom. Collectively, these results indicated that propolis treatment alleviates the alcohol-induced injury of intestinal permeability in mice.

### 3.7. Effect of Propolis on the LKB1/AMPK Signaling Pathway

The LKB1/AMPK signaling pathway is closely associated with the intestinal epithelial tight junctions. Hence, to understand the mechanism underlying the protective effects of propolis against alcohol-induced intestinal mucosal injury in depression mice, we determined the protein expression of RhoA, and the phosphorylation of LKB1 and AMPK in the jejunum and colon tissue. As shown in Figure 5 and Figure 6, the levels of RhoA in mice of the model group were higher than those in mice of the control group, and propolis treatment resulted in obvious downregulations of RhoA (*p* < 0.05). In addition, Western blotting also showed that alcohol exposure down-regulated the expression of p-AMPK and p-LKB1. However, after dietary propolis supplementation, the phosphorylation levels of both AMPK and LKB1 were improved (*p* < 0.05).

### 3.8. Propolis Reduces the Alcohol-Induced Upregulated LPS and FABP2 Levels in the Serum of Mice

To further determine the effect of propolis on intestinal leakage, we investigated the serum levels of LPS and FABP2, which were used as biomarkers of intestinal barrier integrity by an ELISA, as shown in Figure 7A,B. As expected, alcohol exposure increases the serum levels of LPS and FABP2 compared to the control group (*p* < 0.05), while propolis treatment significantly reduced the change levels of LPS and FABP2 caused by alcohol (*p* < 0.05), suggesting the repair effect of propolis on intestinal leakage.

### 3.9. Propolis Attenuates Alcohol-Induced Inflammatory Cytokine Release in the Spleen of Mice

Alcohol exposure was able to increase the levels of different types of pro-inflammatory factors, such as IL-1β, IL-6, and TNF-α, as previously reported [34]. Our ELISA revealed that propolis supplementation (treated group) significantly reduced the levels of these cytokines (*p* < 0.05; Figure 7C–E), which suggested that propolis was able to ameliorate the inflammatory reaction in mice exposed to alcohol.

## 4. Discussion

Increasing epidemiological surveys have demonstrated that chronic alcoholics will be accompanied by a greater risk of major depression, and that higher levels of alcohol use are associated with higher levels of depressive symptoms [42] In the current study, a reduced alcohol intake has been shown to be an effective way to alleviate alcoholic depression, whereas abstinence is still a challenging problem due to alcohol addiction [9] Here, the improvement effect of propolis on alcohol-induced depression-like behaviors in C57BL/6J mice was investigated. Our results demonstrated that propolis intervention was able to reduce the inflammation in the brain gut axis and play a crucial role in neuroprotection.

Generally, the neuropharmacology effects of propolis, on conditions such as neurological disturbance, cerebral ischemia, convulsion, neuroinflammation and cognitive impairment, have always been the focus of attention [43]. However, the studies on the usage of propolis for psychiatric disorders, especially depression have been poorly explored. In the current study, we found that the dietary supplementation of propolis significantly alleviated alcohol-induced depressive behavioral deficits in mice. Forced swim and sucrose preference tests were used to evaluate the depressive symptoms of mice, as in our previous experiments [34]. Compared with the control group, alcohol-induced model mice had increased behavioral despair in the forced swim test, and also had a decreased sucrose consumption rate, a measure of anhedonia, which indicated that the depression model had been successfully constructed. Propolis supplementation was able to effectively alleviate these behavioral changes, revealing preventive and protective effects against alcohol-induced depression. Moreover, propolis administration also significantly attenuated the anxiety and spatial recognition memory impairment associated with depression, through maze tests [44,45].

The hippocampus is an important mood-regulating area of the brain, and is susceptible to stress and depression owing to the high levels of glucocorticoid receptors and glutamate [46]. The neurogenic hypothesis of depression also points out that impaired hippocampal neurogenesis aggravates the development of depression; an increased number of newborn neurons in the brain was able to reverse the process [47,48]. Additionally, accumulating evidence indicates that hippocampal dysfunction, induced by ethanol, is responsible for cognitive loss and depression [49]. Similarly, our results revealed that alcohol consumption contributed to the degeneration and necrosis of cells in the hippocampal CA3 region, which suggested that the hippocampus was associated with alcoholic depression. Previous studies have shown that hippocampal repair is one of the important targets of antidepressant drugs [50,51]. In this study, we found that propolis supplementation improved the pathological changes in hippocampal nerve cells and significantly reduced the denatured cell index in mice with alcohol-induced depression. Thus, propolis treatment effectively protects against alcoholic damage to the hippocampus. BDNF is widely accepted as being critical for learning and memory, cognitive function and mood regulation, and the expression of mature BDNF peptides in the hippocampus has been proven to be associated with depression [52,53,54]. Recently, studies have shown that BDNF expressed in the DA circuit plays an important role in the development of depressive-like behaviors [55,56]. Our previous research has also shown that alcohol exposure affected the depression-like behaviors of mice by altering the expression of BDNF [34]. Hence, we detected the effect of propolis treatment on the expression level of BDNF and DA in alcoholic depression mice. As expected, propolis supplementation significantly improved the level of the two crucial factors closely related to depression.

Nowadays, growing evidence highlights the brain gut axis as a novel strategy for the prevention and treatment of psychiatric disorders, including anxiety and depression [57]. The brain gut axis is a bidirectional neural processing of information between the brain and gut, which is connected by various signaling pathways and plays an essential role in the regulation of maintaining the homeostasis of the CNS [57,58]. Of the numerous studies on the brain gut axis, the importance of the microbial community structure in maintaining mental health has long been appreciated [57,59,60]. Our previous studies have shown that propolis treatment could modify dysbacteriosis and restore the homeostasis of the gut microbiota microecology [30]. However, in parallel, brain gut dysfunction caused by intestinal barrier function injury could also impair one or more pathways, and eventually lead to depression [61]. In this study, we explored the barrier function of the jejunum and colon epithelial cells by analyzing the changes in intestinal epithelial cell junctions and intestinal permeability. These junctions, which connect apical intestinal epithelial cells seal the space between enterocytes and form the major component to maintain the integrity and tightness of the intestinal mucosal barrier [62,63]. The intestinal barrier integrity was damaged in the mice after alcohol exposure, but after intervention with propolis, the impaired intestinal mucosal barrier was effectively improved, with the narrowing of the intercellular spaces in the tight junctions of the intestinal epithelium and alleviated intestinal leakage symptoms. In addition, we also detected the expression of crucial proteins of tight junctions. Western blotting and immunofluorescence showed that propolis treatment increased the reduced protein expression levels of Claudin-1, Occludin and ZO-1 caused by alcohol exposure. Furthermore, propolis treatment also significantly improved the level of intestinal FABP2, which was regarded as a kind of biomarker for intestinal epithelium paracellular integrity molecules [61]. Generally, the translocation of gut-derived endotoxins following intestinal barrier dysfunction and subsequent systemic inflammatory response might be the key event contributing to alcohol-induced depression [61,64]. Since our results showed the repair function of propolis on intestinal barrier function, we further measured the serum levels of LPS in different groups. Obviously, the serum LPS level in alcoholic depression mice was higher than that in the control mice, while the propolis treatment reversed this change, presumably due to the improvement of intestinal leakage. The spleen, the central organ in regulating the inflammation-related immune response, has been supposed to be closely related to the depression-like phenotype induced by LPS [65,66]. Several studies have also shown that the inflammatory response of the spleen is associated with the function of the brain and gut [67,68], implying its important role in the brain gut axis. Therefore, we further detected the levels of pro-inflammatory cytokines in the spleen. As expected, propolis treatment significantly reduced the expression of IL-6, IL-18 and TNF-α in the intervention group compared with that in alcoholic depression mice. Collectively, propolis supplementation was able to ameliorate the brain gut dysfunction of the alcoholic depression mice by reducing the inflammatory response.

To further dissect the role of propolis in the intestinal mucosal injury of alcohol-induced depression mice, we detected the molecules and signal pathways proteins closely related with the intestinal epithelial TJs. Recent studies have demonstrated that RhoA kinase participates in alcohol-induced intestinal barrier disruption through the assembly of TJs and the organization of the actomyosin ring in the polarized epithelia [69,70,71]. According to our experimental results, alcohol exposure increased the protein expression of RhoA kinase, which further verified the above conclusions again. However, after supplementation with propolis, the expression of RhoA kinase was down regulated, hinting the function recovery of intestinal epithelial tight junctions. Of the downstream effectors of RhoA, the phosphorylation of AMPK has been found to be implicated in the construction of TJs by enhancing ZO-1 and Occludin expression [72,73], and protected the epithelial barrier against various environmental stresses [74]. Moreover, the activation of AMPK was able to maintain the stability of TJs in the colon and inhibit the progression of colitis [75]. The LKB1, a major upstream regulated kinase of AMPK subunits [76], is found to trigger and modulate multiple biological processes, such as the inflammatory reaction and oxidative stress, by mediating the LKB1/AMPK signal [77,78,79]. Our results indicated that propolis treatment significantly reversed the decreased phosphorylation of LKB1 and AMPK caused by alcohol exposure in the jejunum and colon. The effect of the regulation of propolis on the protein expression of RhoA kinase and the LKB1/AMPK signal provides further evidence that propolis was able to ameliorate brain gut dysfunction.

## 5. Conclusions

In summary, this study highlights the improvement effect of propolis dietary supplementation on alcohol-induced depressive symptoms by repairing the intestinal mucosal barrier and hippocampal injury, and further improved brain gut dysfunction. These findings provide novel insights into the development of propolis as an alternative functional nutrient for the treatment of alcoholic depression.

## Figures and Tables

**Figure 1 nutrients-14-01213-f001:**
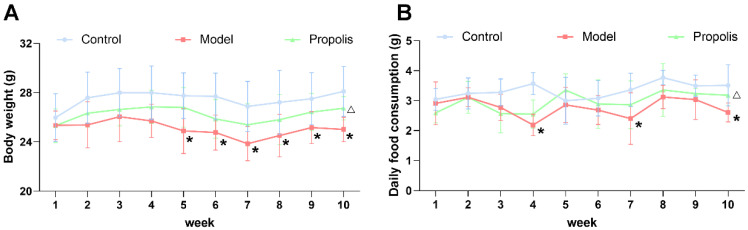
Body weight and daily food consumption in mice within 10 weeks. The average body weight (**A**) and average daily food consumption (**B**) of the mice in the three different groups. * Denotes *p* < 0.05 vs. control group; ^△^
*p* < 0.05 vs. model group.

**Figure 2 nutrients-14-01213-f002:**
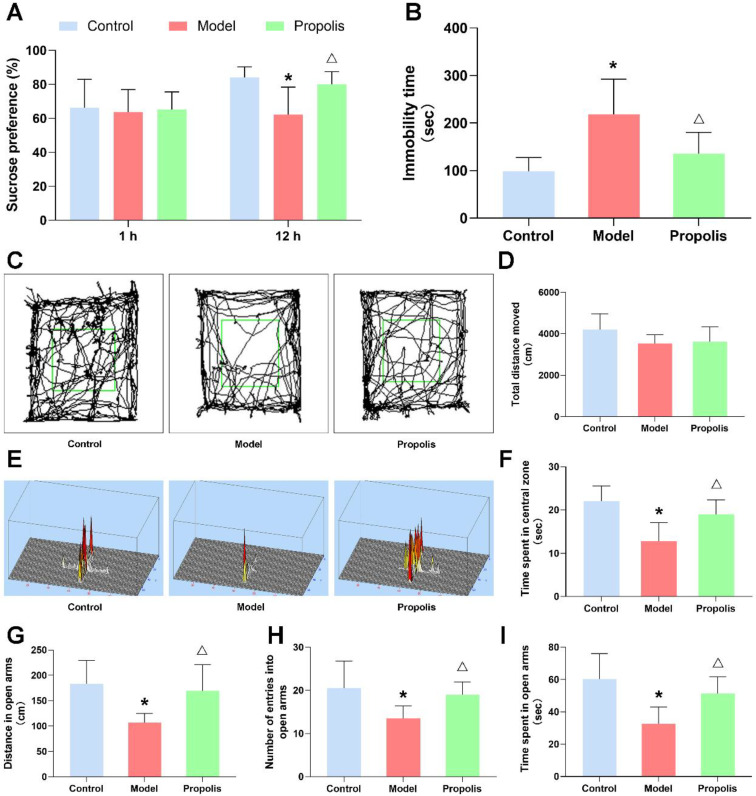
The behavioral performance of mice. (**A**) Sucrose preference of the mice in the different groups. (**B**) Immobility time in the FST. (**C**) Representative animal track in the OFT. Green square area represents the central zone. (**D**) Total distance of mice moved in the OFT. (**E**) Representative activity heat maps in the EPM. (**F**) The locomotion time in the central zone in the OFT. (**G**) Total distance of mice in the open arms of the EPM. (**H**) The number of entries into the open arms of the EPM. (**I**) The locomotion time in the open arms of the EPM. * Denotes *p* < 0.05 vs. control group; ^△^
*p* < 0.05 vs. model group.

**Figure 3 nutrients-14-01213-f003:**
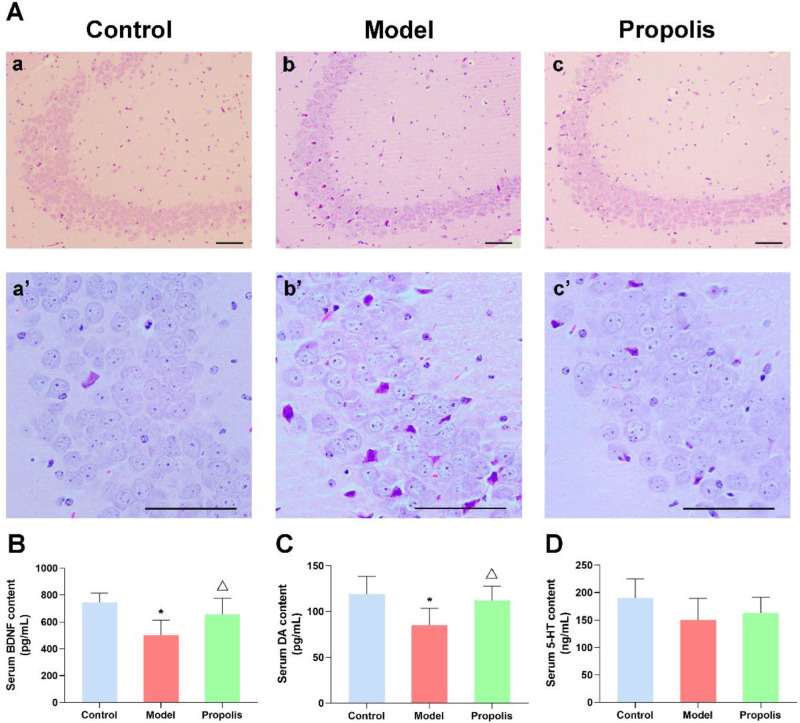
Effects of propolis on alcohol-induced hippocampal injury, BDNF and serum monoamine expression in mice. (**A**) The morphological structure of the hippocampal CA3 region. (**a**,**a’**) control group; (**b**,**b’**) model group; and (**c**,**c’**) propolis group, Scale bars = 200 µm. (**B**) Levels of serum BDNF of the mice in the different groups. (**C**) Levels of serum DA of the mice in the different groups. (**D**) Levels of serum 5-HT of the mice in the different groups. * Denotes *p* < 0.05 vs. control group; ^△^
*p* < 0.05 vs. model group.

**Figure 4 nutrients-14-01213-f004:**
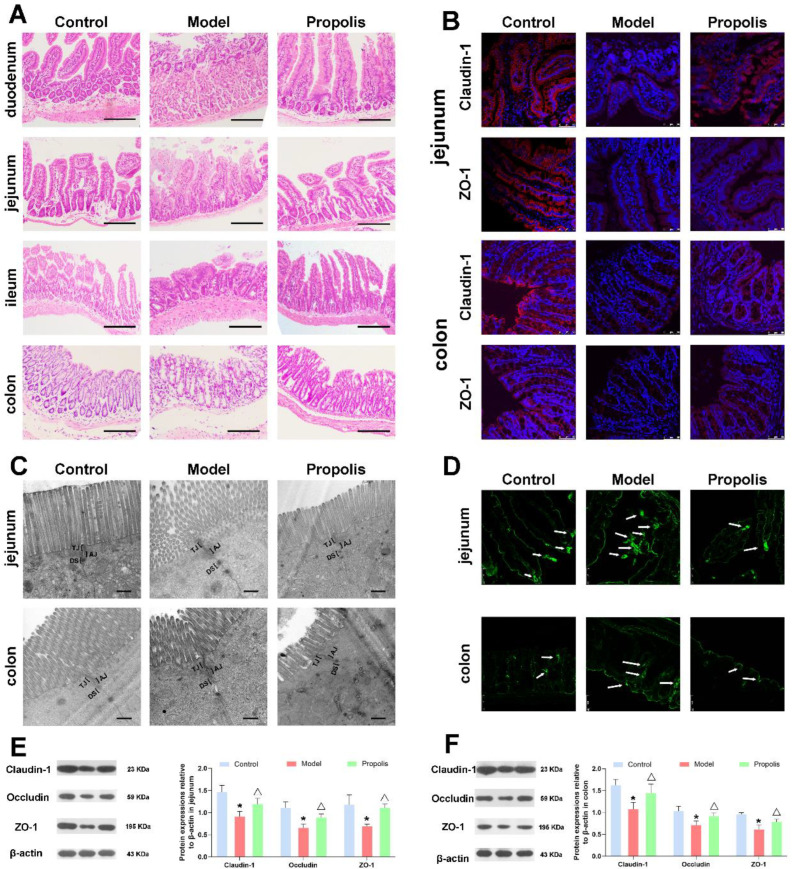
Effects of propolis on intestinal mucosal barrier function. (**A**) Pathological changes of duodenum, jejunum, ileum and colon according to HE staining. Scale bar = 200 µm. (**B**) Immunofluorescence detection of tight junctions related proteins Claudin-1 (red fluorescence) and ZO-1 (red fluorescence) in tissues of the jejunum and colon. Nuclei were counterstained with DAPI (blue fluorescence). Scale bar = 50 µm. (**C**) The ultrastructure of jejunum and colon was observed by transmission electron microscope. Scale bar = 500 nm. (**D**) Biotin tracer assay in jejunum and colon. The green fluorescence signals indicate biotin distribution; the white arrows indicate tracer leakage. Scale bar = 50 μm. (**E**,**F**) Expressions of Claudin-1, Occludin, and ZO-1 proteins in jejunum (**E**) and colon (**F**) examined by Western blotting. * Denotes *p* < 0.05 vs. control group; ^△^
*p* < 0.05 vs. model group.

**Figure 5 nutrients-14-01213-f005:**
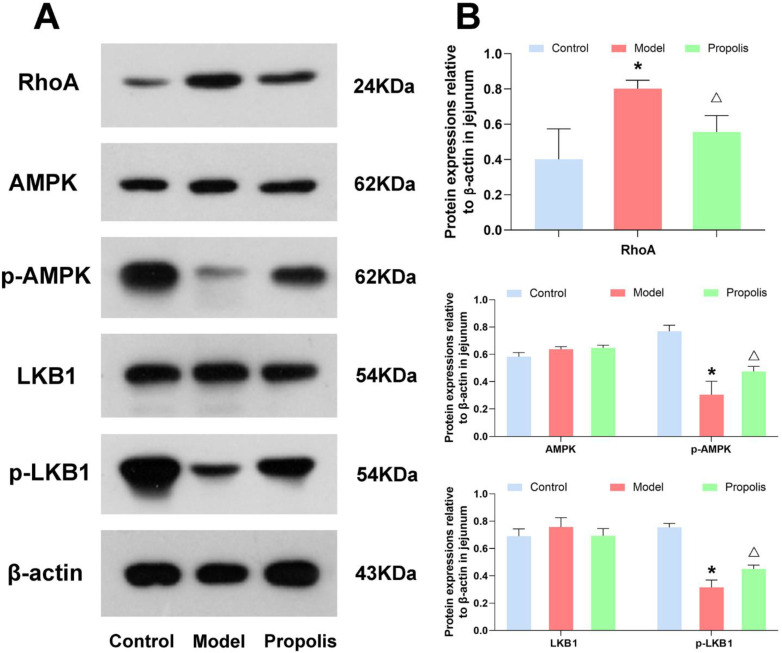
Effects of propolis on the expressions of the LKB1/AMPK signaling pathway in the jejunal tissue determined. (**A**) Western blotting analysis of the expression level of RhoA, AMPK, p-AMPK, LKB1 and p-LKB1 in jejunal tissues. (**B**) Quantitative analysis of the western blotting data. * Denotes *p* < 0.05 vs. control group; ^△^
*p* < 0.05 vs. model group.

**Figure 6 nutrients-14-01213-f006:**
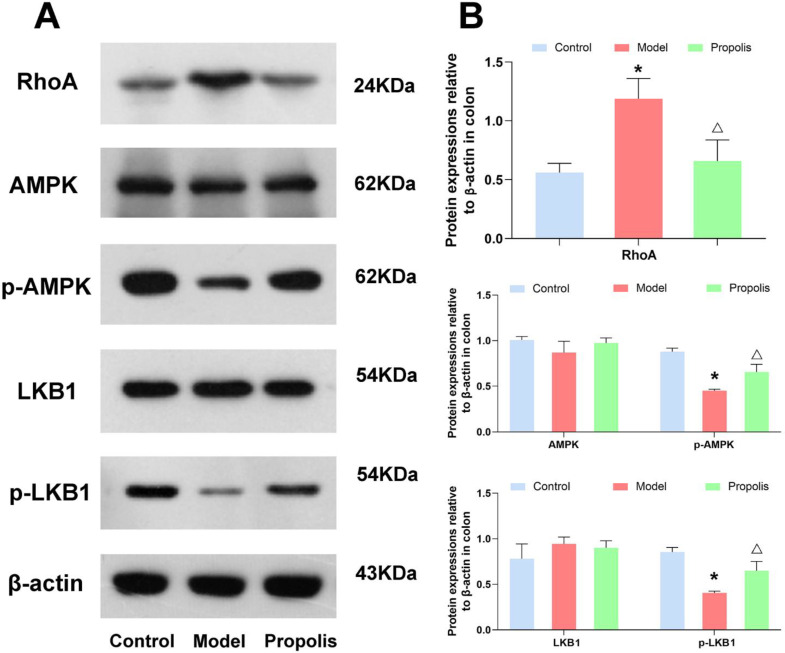
Effects of propolis on the expressions of LKB1/AMPK signaling pathway in the colon tissue. (**A**) Western blotting analysis of the expression level of RhoA, AMPK, p-AMPK, LKB1 and p-LKB1 in colon tissues. (**B**) Quantitative analysis of the western blotting data. * Denotes *p* < 0.05 vs. control group; ^△^
*p* < 0.05 vs. model group.

**Figure 7 nutrients-14-01213-f007:**
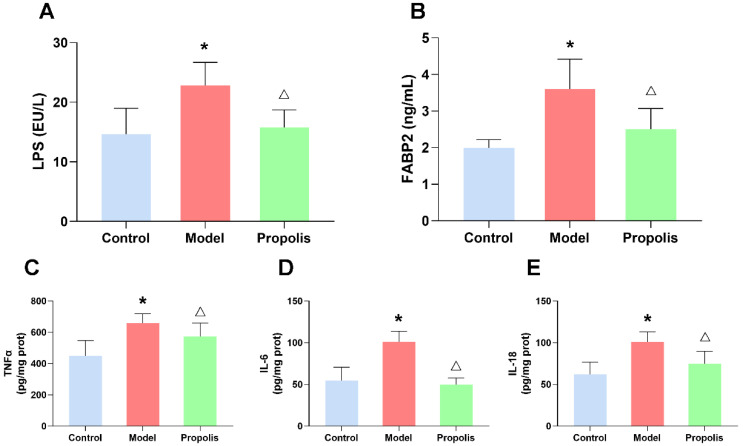
Effects of propolis on alcohol-induced inflammatory reaction. (**A**,**B**) The serum levels of LPS (**A**) and FABP2 (B). (**C**–**E**) The levels of inflammatory cytokines in the spleen. TNF-α (**C**), IL-6 (**D**) and IL-18 (**E**). * Denotes *p* < 0.05 vs. control group; ^△^
*p* < 0.05 vs. model group.

**Table 1 nutrients-14-01213-t001:** The analysis of denatured cell index (DCI) in HE staining. * Denotes *p* < 0.05 vs. control group; ^△^
*p* < 0.05 vs. model group.

Group	n	DCI
Control	6	0.03 ± 0.05
Model	6	0.11 ± 0.07 *
Propolis	6	0.04 ± 0.05 ^△^

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
