# Peer review of "Propolis Ameliorates Alcohol-Induced Depressive Symptoms in C57BL/6J Mice by Regulating Intestinal Mucosal Barrier Function and Inflammatory Reaction"

_nutrients, 2022, doi:10.3390/nu14061213_

Round 1

Reviewer 1 Report

The paper addresses the  effects of propolis on alcohol-induced depressive symptoms in C57BL/6J mice by regulating intestinal mucosal barrier function and inflammatory reaction.

The work in general is interesting, but the results need to be better described and discussed.

The following aspects should be taken into account in order to improve the quality of the paper:

  1. Extensive editing of English language and style required.
  2. The references must be adjusted to the format of the Journal (MDPI)
  3. The abstract must be improved.
  4. Some abbreviations need to be reviewed and described in the text.
  5. Introduction need to reformulated and improved.
  6. Line 45-46: Please explain this sentence. "Excessive alcohol consumption has been suggested to serve as a modulator of inflammatory response"
  7. Line 74-75. Please improve this sentence.
  8. Line 75: "aqueous ethanol extracts" replace with "hydroethanolic extracts" (throughout the document)
  9. please briefly describe extraction conditions (section 2.1), and explain this sentence: ...", followed by stored at 4 °C in the no-light conditions and warmed to room temperature until use" (line 78).
  10. Why are animals housed individually?
  11. Why was all the liquid supply cut off on friday and saturday? after 4 days of alcohol solution? Has this protocol been approved by an ethical committee?
  12. The methodologies need to be rewritten for better understanding of potential readers.
  13. Please note that in the results of table 1 the SD values are higher than the averages. This makes the results unfeasible.
  14. The statistical analysis must be reviewed, considering the SD values (not only the averages). For all results.
  15. The results should be described and discussed according to the statistical results.
  16. The conclusions must be improved.

Reviewer 2 Report

The paper seems very interesting beginning with the title. But reading the manuscript different flaws in writing and description. 

When I wanted to check different methods from the reference list, I did not find the respective studies online, for example, the study of Zhang H et al (for the preparation of propolis). The reference list is not written according to Guide for authors of Nutrients Journal, although no imposed format is necessary, but consistency in writing. I have noticed a CN as Journal name, a Rev. Clin. Psychol, and a The American Journal of Medicine ... This is not consistent. In my opinion, the title of the paper must be inserted no matter what format is used, also doi number. A reviewer is not obliged to search on the internet, where the mentioned article is found, in order to see if the author was using it properly. The methods are not properly described, in order to be reproducible. A list of abbreviations is necessary, or when is used for the first time, a medical term written as an abbreviation must be written: "in extenso".  The discussion section is too large and refers mainly to literature studies, only a small sentence at the end of a long paragraph refers to "our study demonstrates ...."

Round 2

Reviewer 2 Report

The manuscript seems improved, but no changes in the reference list, although the authors specify that this was changed according to Author guidance.  The English language does not seem corrected with the MDPI editor in my opinion. A more clearly expression of the text must be used for all the manuscript.